# In-depth quantitative proteomics uncovers specie-specific metabolic programs in *Leishmania (Viannia)* species

**Nathalia Pinho**[1], **Jacek R. Wiśniewski**[2], **Geovane Dias-Lopes**[3], **Leonardo Saboia-Vahia**[1¤a], **Ana Cristina Souza Bombaça**[4], **Camila Mesquita-Rodrigues**[4], **Rubem Menna-Barreto**[4], **Elisa Cupolillo**[1], **Jose Batista de Jesus**[3,5], **Gabriel Padrón**[1¤b], **Patricia Cuervo**[1]*

**1** Laboratório de Pesquisa em Leishmanioses, Instituto Oswaldo Cruz, Fiocruz, Rio de Janeiro, RJ, Brazil, **2** Biochemical Proteomics Group, Department of Proteomics and Signal Transduction, Max-Planck-Institute of Biochemistry, Martinsried, Germany, **3** Laboratório de Biologia Molecular e Doenças Endêmicas, Instituto Oswaldo Cruz, Fiocruz, Rio de Janeiro, RJ, Brazil, **4** Laboratório de Biologia Celular, Instituto Oswaldo Cruz, Fiocruz, Rio de Janeiro, RJ, Brazil, **5** Departamento de Medicina–Universidade Federal de São João Del Rei, Campus Dom Bosco, São João del Rei, MG, Brazil

¤a Current address: Laboratório Parasitologia Celular e Molecular, Instituto René Rachou, Fiocruz, Belo Horizonte, MG, Brazil
¤b Current address: Center for Genetic Engineering & Biotechnology, La Habana, Cuba
* patricia.cuervo@fiocruz.br

**Data Availability Statement:** All relevant data are within the manuscript and its Supporting Information files. Mass spectrometry data are

## Abstract

*Leishmania* species are responsible for a broad spectrum of diseases, denominated Leishmaniasis, affecting over 12 million people worldwide. During the last decade, there have been impressive efforts for sequencing the genome of most of the pathogenic *Leishmania* spp. as well as hundreds of strains, but large-scale proteomics analyses did not follow these achievements and the *Leishmania* proteome remained mostly uncharacterized. Here, we report a comprehensive comparative study of the proteomes of strains representing *L. braziliensis*, *L. panamensis* and *L. guyanensis* species. Proteins extracted by SDS-mediated lysis were processed following the multi-enzyme digestion-filter aided sample preparation (FASP) procedure and analysed by high accuracy mass spectrometry. "Total Protein Approach" and "Proteomic Ruler" were applied for absolute quantification of proteins. Principal component analysis demonstrated very high reproducibility among biological replicates and a very clear differentiation of the three species. Our dataset comprises near 7000 proteins, representing the most complete *Leishmania* proteome yet known, and provides a comprehensive quantitative picture of the proteomes of the three species in terms of protein concentration and copy numbers. Analysis of the abundance of proteins from the major energy metabolic processes allow us to highlight remarkably differences among the species and suggest that these parasites depend on distinct energy substrates to obtain ATP. Whereas *L. braziliensis* relies the more on glycolysis, *L. panamensis* and *L. guyanensis* seem to depend mainly on mitochondrial respiration. These results were confirmed by biochemical assays showing opposite profiles for glucose uptake and $O_2$ consumption in these species. In addition, we provide quantitative data about different membrane proteins, transporters, and lipids, all of which contribute for significant species-specific differences and

available via ProteomeXchange with identifier PXD017696.

**Funding:** This work was supported by the Conselho Nacional de Desenvolvimento Científico e Tecnológico - CNPq (P.C. Universal grant No. 423300/2018-0); FIOCRUZ (P.C., N.P, L.S.V, E.C, G.P. PAEF grant No. IOC-023-FIO-18-2-63); Fundação de Amparo à Pesquisa do Estado de Rio de Janeiro - FAPERJ (P.C. JCNE E-26/201.545/ 2014 and JCNE E-26/203.253/2017; N.P TCT No. E-26/202.464/2017; E.C. CNE E26-202.569/2019); Max-Planck Society for the Advancement of Science (JRW), and Coordenação de Aperfeiçoamento de Pessoal de Nível Superior – CAPES - Finance Code 001 (P.C Process No. 88887.374332/2019-00). G.P. was a CAPES fellow of the Visitant Professor Program (Process No. 0344141). P.C, E.C. and J.B.J. are CNPq Researchers fellows (P.C. Process No. 306393/ 2014-0 and 305796/2017-8; E.C. Process No. 302622/2017-9; J.B.J Process No. 309732/2018-2). The funders had no role in study design, data collection and analysis, decision to publish, or preparation of the manuscript.

**Competing interests:** The authors have declared that no competing interests exist.

provide rich substrate for explore new molecules for diagnosing purposes. Data are available via ProteomeXchange with identifier PXD017696.

## Author summary

*Leishmania braziliensis*, *L. panamensis*, and *L. guyanensis* are responsible for most of the cases of tegumentary leishmaniasis (TL) in the Americas. These species are associated with a variety of clinical manifestations of TL ranging from self-healing localized cutaneous lesions to disseminated and mucocutaneous presentations that may result in severe oropharyngeal mutilation. Here, we report a comprehensive quantitative comparison of the proteome of those species. Assessment of absolute titers of ~7000 proteins revealed a very clear differentiation among them. Significant differences in energy metabolism, membrane proteins, transporters, and lipids are contributing for species-specific traits and provide rich substrate for exploring new molecules for diagnosing purposes.

## Introduction

*Leishmania* species are responsible for a broad spectrum of diseases denominated Leishmaniasis that affects over 12 million people worldwide [1]. *Leishmania braziliensis*, *L. panamensis*, and *L. guyanensis* are species of the *Leishmania* (*Viannia*) subgenus that comprises exclusively Neotropical parasites currently distributed across the American continent. These species are responsible for most of the cases of localized cutaneous leishmaniasis (LCL) in the Americas, but are also associated with the mucocutaneous form of the disease [1]. Mucocutaneous leishmaniasis (MCL) results from the parasite metastasis to the oropharyngeal mucosal tissues via lymphatic or haematogenous dissemination. In advanced stages, MCL may result in severe mutilation by destruction of the palate and nose cartilage [2]. Moreover, whereas *L. panamensis* is mainly associated to LCL, *L. braziliensis* may cause multiple disseminated cutaneous lesions, frequently refractory to treatment [1,3]. In addition, in some regions *L. guyanensis* is frequently isolated from patients who failed to response to pentavalent antimonial (Sb$^V$), the first line treatment for dermal leishmaniasis [4]. Despite that severity, currently there are no prognostic tools for the prediction of the clinical outcome after infection with those species, neither proper species-specific diagnostic tools that may orientate the therapeutic decision.

Heterogeneous clinical manifestations are the result of a complex interaction among host immune responses, host genetic background and parasite characteristics such as virulence, infectivity and pathogenicity [5]. Among others, to understand the parasite contribution for the clinical outcome, complete genome sequences of most of the pathogenic *Leishmania* species and hundreds of strains have been obtained [6–17], paving the way for the analyses of their proteomes. However, such a massive effort has not been followed by proteomics studies.

The first more comprehensive proteomics identification of a *Leishmania* species was accomplished using gel-free analysis of *L. donovani* and *L. mexicana*, with up to ~1800 proteins identified, representing ~20% of the *Leishmania* predicted proteome [18,19]. In addition, protein abundance was estimated for the proteins identified in the *L. mexicana* dataset by calculating the exponentially modified protein abundance index (emPAI) [19]. Recently, proteogenomic re-analysis of *L. donovani* proteome identified ~4000 unique proteins [20]; studies on the *L. mexicana* flagellar proteome identified ~2800 proteins, among which, 701 composed the flagellar proteome [21]; subcellular multi-step fractionation of *L. donovani* allowed

identification of ~3900 proteins, included 645 integral membrane proteins [22]. Despite those important achievements, no *Leishmania* proteome has been described in large-scale nor deep quantitative data is available.

Seminal works on *L. (Viannia)* proteomes used 2DE of cell lysates with the aim to differentiate distinct *Leishmania* species causing American tegumentary leishmaniasis [23]. Since that, advances in MS protein identification driven the first proteomic maps of *L. braziliensis*, *L. panamensis* and *L. guyanensis* [24–35]. Using different methods for sample preparation, up to some hundreds of proteins were identified in these species. Regardless these efforts, identification and quantitative organization of the *L. (Viannia)* species proteome remains mostly uncharacterized.

Despite the advances in mass spectrometry instrumentation, the quality of the sample preparation still determines the quality and depth of the proteomic analysis. The combination of SDS as denaturant for efficient protein extraction, followed by depletion of the detergent by successive washes with concentrated urea in a filter device and consecutive digestion with LysC and trypsin has proven to be an efficient method for sample preparation [31–33].

Here, we describe a comprehensive quantitative analysis of the proteome of three strains that represent distinct and clinically/epidemiologically relevant species of *L. (Viannia)* subgenus: *L. braziliensis* (strain MHOM/BR/2000/LTCP 13396), *L. panamensis* (strain MHOM/CO/2009/6634) and *L. guyanensis* (strain MHOM/BR/1997/NMT-MAO 292P). We combined the filter-aided sample preparation (FASP) method with the consecutive multi-enzyme digestion [31,32] followed by high accuracy mass spectrometry and the "Total Protein Approach" and "Proteomic Ruler" [34,35] for calculation of absolute protein abundances in those species. Our dataset comprises near 7000 proteins, representing the most complete *Leishmania* proteome yet reported, and provides the first comprehensive quantitative analysis of the proteomes of the three species in terms of protein concentration and copy numbers. We describe the basic subcellular organization of the parasites and the cumulative abundance of proteins from the major metabolic processes and demonstrate the main differences among those three species.

## Methods

### Parasite culture

*Leishmania braziliensis* (MHOM/BR/2000/LTCP 13396), *L. panamensis* (MHOM/CO/2009/6634) and *L. guyanensis* (MHOM/BR/1997/NMT-MAO 292P) strains used in this study are open deposits and were provided by the Collection of *Leishmania* of the Oswaldo Cruz Foundation, CLIOC (http://clioc.fiocruz.br/). Through the text, tables and figures those strains will be mentioned as *L. braziliensis*, *L. panamensis* and *L. guyanensis*. Parasites were grown in Schneider's medium containing 20% fetal bovine serum (heat inactivated at 56°C for 50 min) at 25°C. Parasites were collected in the late logarithmic growth phase by centrifugation at 1800 x g for 5 min, and washed twice in PBS, pH 7.2. In order to obtain meaningful data, we analyzed parasites collected from five independent biological assays (five biological replicates) of each *L. braziliensis* and *L. panamensis* and four independent biological assays (four biological replicates) of *L. guyanensis*. Variables such as the lots of culture medium and serum were controlled. According with the Brazilian Law of Biodiversity, this study was registered at SisGen (AA2236F).

### Sample preparation

Parasites were lysed in a buffer containing 0.05 M Tris-HCl, pH 7.6, 0.05 M DTT and 2% SDS in a boiling water bath for 5 min. After chilling to room temperature, the lysates were clarified by centrifugation at 10,000×g for 5 min. Total protein was determined by BCA using a Nanodrop instrument. Aliquots containing 100 μg total protein were processed by the MED-FASP (Multi-Enzyme Digestion—Filter Aided Sample Preparation) method using consecutively

LysC and trypsin [32,33]. Peptides were collected, concentrated and desalted on a C18 reverse phase column.

## Liquid Chromatography–Tandem Mass Spectrometry

The peptide mixtures were analysed using a Q-Exactive HF mass spectrometer (Thermo-Fisher Scientific, Palo Alto). The chromatography was performed using 2 µg of total peptide on a 50 cm column with 75µm inner diameter packed with 1.8 µm $C_{18}$ particles (100 Å pore size; Dr. Maisch, Ammerbuch-Entringen, Germany). Peptide separation was carried out using a 105 min acetonitrile gradient in 0.1% formic acid at a flow rate of 250 nl/min. The backpressure varied between 450 and 650 bar. The Q-Exactive HF mass spectrometer was operated in data-dependent mode with survey scans acquired at a resolution of 50 000 at $m/z$ 400 (transient time 256 ms). Up to the top 15 most abundant isotope patterns with charge $\geq$ +2 from the survey scan (300–1650 $m/z$) were selected with an isolation window of 1.6 $m/z$ and fragmented by HCD with normalized collision energies of 25. The maximum ion injection times for the survey scan and the MS/MS scans were 20 and 60 ms, respectively. The ion target value for MS1 and MS2 scan modes was set to $3 \times 10^6$ and $1 \times 10^5$, respectively. The dynamic exclusion was 25 s and 10 ppm.

## Proteomic data analysis

The data were analyzed with MaxQuant Software using the *L. braziliensis* protein sequences available at UniProt database. The parameters of the search included a fragment ion mass tolerance of 0.5 Da and parent ion tolerance of 20 ppm. The maximum false peptide and protein discovery rate was set as 1%. Cysteine carbamidomethylation was set as fixed modification, methionine oxidation as variable modification, and a maximum of two missed cleavages was allowed. Protein abundances were calculated based on the spectral protein intensity (raw intensities) using the 'Total Protein Approach' (TPA) for absolute protein quantification and the 'Proteomic Ruler' approach [35] for assessing the protein copy number per cell, total protein, protein concentration and copy number were calculated in Microsoft Excel. The Perseus software (Ver. 1.6.5.0) [36] was used for statistical analysis. Minimal number of valid values was set to 4 per protein, for at least one species, and missing values were imputed from a normal distribution. Significances were calculated using the *t*-test with a threshold of false discovery rate (FDR) of 1%. The data are presented as the mean ± SD.

## Enrichment analysis based on Gene Ontology and Metabolic Pathway annotations

In order to analyze if there were particular groups of proteins that contributed the most for distinguishing the *L.* (*Viannia*) species, only the proteins that presented simultaneously significant differences in concentration among the three species were used for enrichment analysis of gene ontology and metabolic pathway annotations using the enrichment tool at Tritrypdb (http://tritrypdb.org, v46). Parameters were set as follows: organism, *L. braziliensis* MHOM/BR/75/M2904 and p-value cutoff, 0.05.

## Analysis of 2-NBDG uptake

Procyclic promastigotes of the late logarithmic growth phase ($5\times10^6$ parasites/mL) were washed (2000 x g for 10 min) and pre-incubated in 10 mM phosphate buffered saline (PBS, pH 7.2, Sigma-Aldrich, St. Louis, USA) for 30 min at 28˚ C. Then, the cells were incubated with 300 µM 2-(N-(7-Nitrobenz-2-oxa-1,3-diazol-4-yl)Amino)-2-Deoxyglucose (2-NBDG, Molecular Probes, Massachusetts, USA), a fluorescent glucose analogue, for 30 min at 28˚ C. The

2-NBDG specificity was monitored by the parasite incubation at 4˚C for 30 min, to decrease the glucose uptake. After this time, the parasites were washed and resuspended in PBS and 2-NBDG+ promastigotes were analyzed using a CytekDxP Multi-Color Upgrades flow cytometer (Cytek, Fremont, USA). A total of 10,000 events were acquired in the region previously established to correspond to protozoa and analyses were performed in Summit 6.1 software (Beckman Coulter, Brea, USA). Three biological replicates were analyzed. Student´s *t* test was used for analyzing the statistical significance of the differences observed among species, values of $p < 0.05$ were accepted and data are presented as the mean ± SD.

### Analysis of oxygen uptake

Promastigotes ($2.5 \times 10^7$ cells/mL) were resuspended in 2 mL of respiration buffer (125 mM sucrose, 65 mM KCl, 10 mM Tris-HCl pH 7.2, 1 mM $MgCl_2$, and 2.5 mM potassium phosphate monobasic) at 26˚C under continuous stirring and analyzed in a high-resolution oxygraph 2K (Oroboros Instruments, Innsbruck, Tyrol, Austria). Mitochondrial oxygen consumption was confirmed by the addition of 2 μM antimycin A (AA) (Sigma-Aldrich, St. Louis, Missouri, USA) to reach residual oxygen consumption (ROX). Oxygen concentration and flux were acquired and analyzed using the DatLab software. Statistical significance was assessed by Student´s *t* test and results are shown as the mean ± SD of three independent experiments (three biological replicates).

### Gene expression analysis

Gene expression levels were analyzed by qPCR for 10 genes encoding for proteins that presented statistical differences of abundance among at least two species. Oligonucleotide sequences of the target and reference genes (S1 Table) were designed using the Primer-BLAST designing tool from available public sequences for *L. braziliensis*. Total RNA was extracted from cells by using Trizol Reagent according to the manufacturer's instruction. The quantity and integrity of RNA were determined by spectrophotometer (Nanodrop ND-1000, Nano-Drop Technologies); cDNA was synthesized from 2 μg of total RNA using SuperScript III reverse transcription system and oligo-dT primers in total reaction volumes of 20 μ*L*. Real-time PCR was carried out in duplicate using SYBR Green PCR Master Mix (Applied Biossystems) and the Vii7 Sequence Detection System (Applied Biossystems), following the manufacturer's procedures. PCR conditions for all primers were optimized and specificities were verified by melting curve analyses and agarose gel electrophoresis. The cDNAs were amplified during 40 cycles consisting of 10 s of denaturation at 95˚C, 20 s of annealing at 56˚C, and 20 s of extension at 72˚C. Gene expression was quantified by means of the comparative Ct method (ΔΔCt). The linearity of the assay was evaluated using a standard curve for each gene from serial dilutions of pooled cDNAs from all samples. The efficiency of amplification for each gene was calculated from the standard curve of each gene. Data are shown as normalized ratios between target gene expression and geometric media of the reference genes. Three independent assays were performed following the MIQE guidelines [37].

## Results and discussion

### The *Leishmania* (*Viannia*) proteome is distributed in about six orders of magnitude and *L. braziliensis*, *L. panamensis* and *L. guyanensis* are clearly separated by differences in their protein concentrations

To obtain an in-depth quantitative description of the *L.* (*Viannia*) proteome, whole cell lysates, of four-five independent biological replicates, from each of the species were processed by

MED-FASP and analyzed by LC-MS/MS. Searches with MaxQuant software resulted in the identification of 120224 peptides corresponding to 6930 proteins, of which 95% (6530) were identified with at least three peptides and 80% (5544) with at least six peptides (S1 Fig, S2 Table). Our dataset comprises ~85% of the *Leishmania* predicted proteome (~8000 protein-coding genes predicted, considering one protein per gene) [6,7,9,12] and therefore represents the most complete proteome for a *Leishmania* species yet described. More than 3000 proteins were identified for the first time (S2 Table). Pearson's correlation analysis revealed high reproducibility of proteomics data between biological replicates (coefficient > 94%) (S1 Fig). More than 6600 proteins were identified in each replicate and 6033 proteins were identified in all 14 samples (S3 Table).

Analysis of the data showed that only 9% of identified proteins have Gene Ontology annotations for cellular component, molecular function and biological process together (S2 Fig). Considering only the ontological annotation for cellular component we found that about 4900 proteins in our data set appear as uncharacterized. Despite efforts for improving *L. braziliensis* annotations [38], functional annotation of *Leishmania* spp. is still very limited and represents a bottleneck in the biological interpretation of transcriptomics and proteomics data.

After statistical analysis using Perseus, a total of 6821 proteins were statistically validated in at least 10 out of the 14 samples (FDR 1%) and these were used for estimation of total protein contents, protein concentrations and protein copy numbers using the total protein approach (TPA) and the histone ruler methods. Principal component analysis (PCA) of the protein concentrations showed tight clustering for each set of biological replicates and revealed a clear separation among the strains representing *L. braziliensis*, *L. panamensis* and *L. guyanensis* samples (Fig 1A). Although proteomes are highly dynamic and may vary according to parasite stage, culture conditions and probably to intra-specific diversity of *Leishmania*, our results show that global protein abundances obtained by deep proteomics are able to clearly distinguish the three species, in a similar way that deep genome sequencing allows such distinction [9,12].

The total protein contents per cell were calculated using the histone ruler method based on the genome data for *L. braziliensis*, *L. panamensis* and *L. guyanensis* (strains M2904, PSC-1 and CL085, respectively) and that report 32Mb, 30Mb and 31Mb DNA, respectively [6,9,12]. These values correspond to 0.035 pg, 0.033 pg and 0.034 pg of DNA, respectively, per haploid genome. Assuming diploidy, *L. braziliensis*, *L. panamensis* and *L. guyanensis* strains used in this study contain 3.86 ± 0.1 pg, 3.89 ± 0.08 pg and 3.55 ± 0.2 pg of total protein per cell, respectively (Fig 1B, Table 1). These values are similar to the protein estimation reported by Aebischer [39] for *L. mexicana*, which corresponds to ~4 pg. Although we are aware that these calculations could vary according to the ploidy of the genome, the ploidy status of the strains here studied is unknown. Because it has been shown that the chromosome number in *Leishmania* sp. (i) varies among species and strains [7,8,12,13]; (ii) varies in response to external stimuli [13,15,16, 40,41]; (iii) is present as a mosaic, *i.e.*, different chromosomes have different ploidy [42]; (iv) is highly variable in vitro-cultivated strains but does not seem to vary as much in intracellular parasites from mammal hosts [17,41], and (v) has been barely studied in species of the subgenus *L. (Viannia)*, we opted for a parsimonious assumption of a diploid state for the three species here studied. Nevertheless, the power of the quantitative method used here can at any time revisit the calculations and adjust the protein concentrations according to the observed ploidy for each chromosome in each species, since it does not depend on chemical markers or external standards.

Using the values of total protein contents, we calculated protein copy numbers per species and estimated the total protein molecules per parasite to be 6.4 ± 0.20 x $10^7$ for *L. braziliensis*, 6.3 ± 0.13 x $10^7$ for *L. panamensis*, and 5.9 ± 0.35 x $10^7$ for *L. guyanensis* (Table 1). The protein

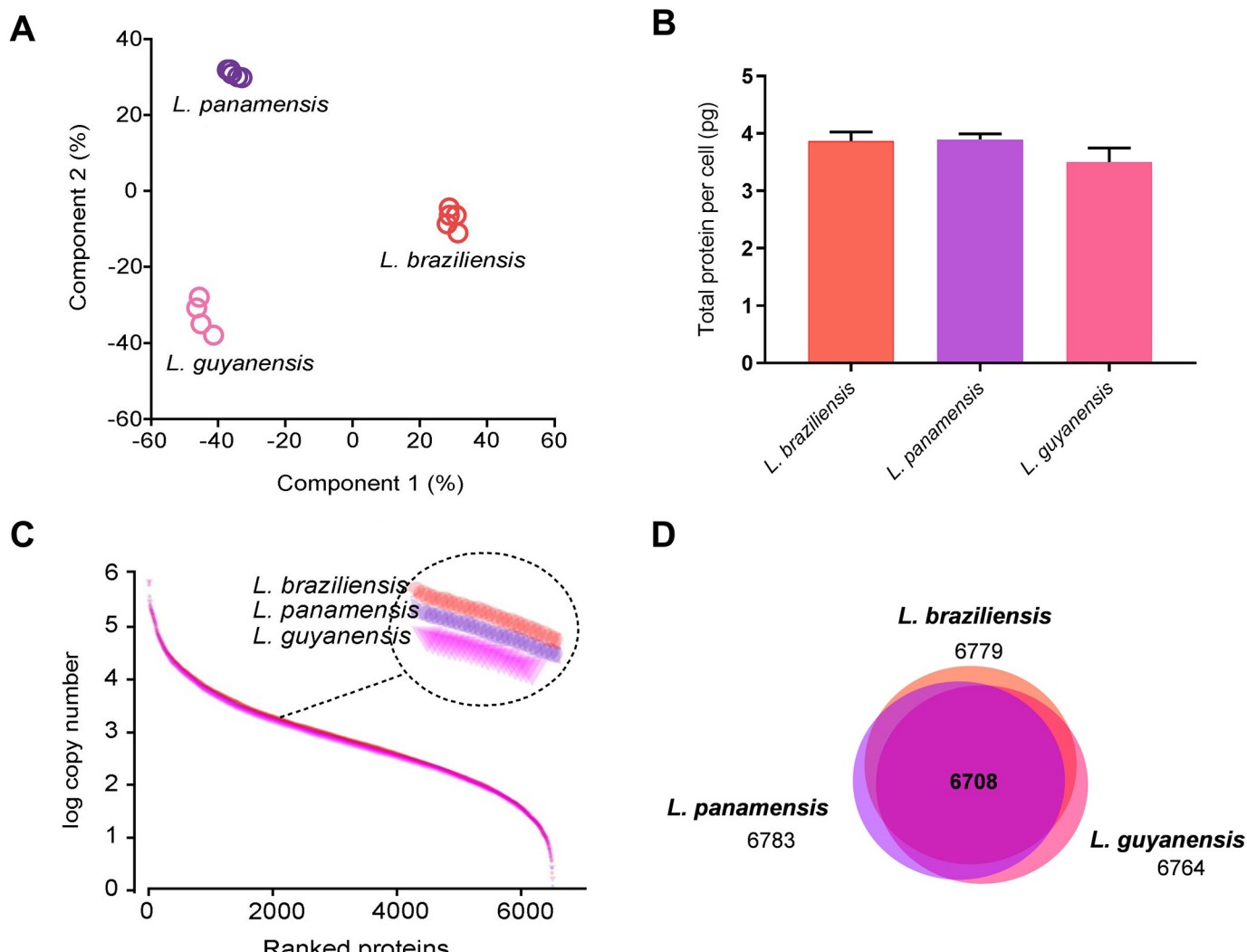

**Fig 1. Description of the *Leishmania* (*Viannia*) proteomes.** Whole cell lysates from *L. braziliensis*, *L. panamensis* and *L. guyanensis* were processed by MED-FASP and analyzed by LC-MS/MS. Protein abundances were calculated based on the raw spectral intensities. **(A)** Principal component analysis of the protein concentration values of all identified proteins determined by the *Total Protein Approach*, (TPA), method. The close clustering of each set of biological replicates reveals accurate reproducibility among independent biological samples. **(B)** Total protein content per cell, n = 5 biological replicates for *L. braziliensis* and *L. panamensis* and n = 4 for *L. guyanensis*. Values are expressed in pg ± SD. **(C)** Dynamic range of protein copy numbers across all proteins identified in each species. Protein abundances were ranked from the highest to the lowest copy number in the *Leishmania* samples. **(D)** Venn plot with number of proteins identified in each species.

**Table 1. Summary of general information about the proteomes of *Leishmania braziliensis*, *L. panamensis* and *L. guyanensis* promastigotes.**

|  | *L. braziliensis* | *L. panamensis* | *L. guyanensis* |
| --- | --- | --- | --- |
| International code | MHOM/BR/2000/LTCP 13396 | MHOM/CO/2009/6634 | MHOM/BR/1997/NMT-MAO 292P |
| Biological replicates | 5 | 5 | 4 |
| Peptides identified | 78398 | 77940 | 62018 |
| Protein groups identified | 6779 | 6783 | 6764 |
| Total protein per cell | 3.86 ± 0.1 pg | 3.89 ± 0.08 pg | 3.55 ± 0.2 pg |
| Total protein molecules per parasite | $6.4 ± 0.20 \times 10^7$ | $6.3 ± 0.13 \times 10^7$ | $5.9 ± 0.35 \times 10^7$ |

copy number values span 6 orders of magnitude (Fig 1C). A total of 6779 proteins were identified in *L. braziliensis*, 6783 in *L. panamensis* and 6764 in *L. guyanensis.* A total of 6708 proteins were identified as common to the three species (Fig 1D).

Among the top 20 most abundant proteins we found histones H3, H4, H2B, alpha- and beta-tubulin, elongation factor 1-alpha, HSP70 and HSP83, among others, with an estimated mean among the three species around 750000, 580000, 680000, 560000, 545000, 530000, 400000, 300000 copies per cell, respectively (Table 2). These proteins are in fact the most abundant in the proteomes of other *Leishmania* species, [43,44] and, probably for this reason, they are frequently identified in proteomic studies independently of the phenomenon studied. Interestingly, those proteins are coded by gene arrays with more than 10 gene copies each one [7,9,12], suggesting a correlation between gene copy number and protein copy number, at least for those genes, in the strains of *L. (Viannia)* species studied here. Among other highly abundant molecules were various ribosomal proteins including several 40S and 60S subunits involved in translation of mRNA. Such molecules were also identified among the top 30 most highly expressed genes in different isolates of *L. tropica* [11]. However, some of the most abundant gene arrays reported to date in *L. braziliensis*, *L. panamensis* and *L. guyanensis*, such as amastin-like surface protein, TATE DNA transposon, GP63, folate/biopterin transporter and tuzin [7,9,12], were not among the most abundant proteins (top 20, Table 2, or even top 100, S2 Table) in our dataset, evidencing that for those proteins there is a poor correlation between gene and protein copy numbers. Nevertheless, such lack of correlation could be associated, among others, with the parasite stage, *i.e.*, some gene arrays are more recruited during

**Table 2. Top 20 most abundant protein groups in *Leishmania* (*Viannia*) proteomes in terms of protein copy number.** Protein IDs according to UniProt. Detailed information in S2 Table. Lb: *L. braziliensis*; Lp: *L. panamensis*; Lg: *L. guyanensis*.

| Majority protein ID | Protein names | Number of proteins | Peptides | Mol. weight [kDa] | Lb Average copy number | Lp Average copy number | Lg Average copy number |
|---|---|---|---|---|---|---|---|
| A4H675 | Histone H3 | 1 | 19 | 14.76 | 683514 | 781749 | 797009 |
| A4H7X1 | Histone H4 | 5 | 17 | 11.36 | 632678 | 518407 | 591973 |
| A4H9W2 | Histone H2B | 2 | 16 | 12.40 | 615245 | 701910 | 721757 |
| A4HLC9 | Tubulin beta chain | 4 | 38 | 49.76 | 596408 | 564945 | 516207 |
| A4H727 | Tubulin alpha chain | 3 | 52 | 49.68 | 566849 | 561107 | 507227 |
| A4H8V4 | Elongation factor 1-alpha | 1 | 51 | 49.07 | 550167 | 489947 | 554094 |
| A4HBV1 | Histone H2A | 3 | 16 | 14.13 | 520962 | 354314 | 320116 |
| A4HMP3 | Kinetoplastid membrane protein-11 | 1 | 17 | 11.17 | 501495 | 545944 | 420273 |
| A4HER9 | Putative 10 kDa heat shock protein | 1 | 12 | 10.75 | 477486 | 294995 | 301477 |
| A4HGY1 | Putative heat-shock protein hsp70 | 3 | 70 | 71.28 | 429433 | 353265 | 400574 |
| A4H5R9 | Putative calmodulin | 1 | 17 | 16.82 | 392799 | 434208 | 399249 |
| A4HGV8 | Putative 40S ribosomal protein S17 | 1 | 21 | 16.56 | 350644 | 286853 | 297001 |
| A4H7N9 | Putative calpain-like cysteine peptidase | 1 | 15 | 13.06 | 339950 | 295024 | 232822 |
| A4HKT8 | Nucleoside diphosphate kinase (EC 2.7.4.6) | 2 | 16 | 16.69 | 336687 | 301436 | 167969 |
| A4HL70 | Heat shock protein 83–1 | 1 | 80 | 80.66 | 327133 | 281794 | 260786 |
| A4H8H2 | Putative 60S ribosomal protein L21 | 2 | 24 | 18.03 | 326347 | 292275 | 307616 |
| A4HD62 | Putative 60S ribosomal protein L17 | 1 | 15 | 19.01 | 316904 | 296723 | 301417 |
| A4HMZ5 | Putative 60S ribosomal subunit protein L31 | 1 | 21 | 21.38 | 301492 | 254432 | 243122 |
| A4HAF8 | Contig, possible fusion of chromosomes 20 and 34 | 1 | 12 | 13.07 | 299331 | 231468 | 250427 |
| A4HGE0 | 40S ribosomal protein S14 | 1 | 15 | 15.67 | 294401 | 248622 | 256815 |

procyclic or metacyclic promastigote stage whereas others should be preferentially recruited during amastigote stage. Indeed, it has been shown that amastin is preferentially expressed by amastigotes during the interaction with macrophages [45]. On the other hand, deep proteome data as that obtained here allowed us to detect, identify and quantify low abundant proteins such as a protein kinase (A4HLK1), a subtilisin-like serine peptidase (A4H792) and a lot of uncharacterized proteins (ex: A4HM75, A4HP44 and A4H8S3), all of them presenting around 50–80 copies per cell (S2 Table). Interestingly, the quantitative data here provided may impact the selection of vaccine candidate antigens. Indeed, because protein abundance will determine its chance to be processed and loaded onto MHC molecules, it has been proposed that protein abundance is the single most important parameter to take into account when a vaccine candidate antigen is being selected [39].

## The protein copy number has low correlation with the protein molecular weight

We analyzed the correlation of the protein abundance in terms of copy number, using the mean value obtained for the three species, against the molecular weight of the identified proteins. Although, in general, smaller proteins present more copies than larger proteins, we observed that there is a low correlation between the number of molecules per protein and the molecular weight ($R^2 = 0.129$) (Fig 2A). For example, the putative calpain-like cysteine peptidase (A4HFH6) with 701.07 kDa, one of the largest proteins in our proteome dataset, presented approximately 29400 copies, representing ~0.05% of total parasite protein mass, in contrast to smaller putative calpain-like cysteine peptidases (A4HJ22 and A4HJ21) with around 100 kDa that presented 200–600 copies per cell. Our data agree with previous reports on *Saccharomyces cerevisiae* and *Escherichia coli* protein abundance that found moderate or no correlation between protein size and protein abundance [46,47].

We also analyzed the protein abundance of basic components of promastigotes architecture and metabolism, in terms of percentage of total cellular protein or copy number. Based on the current gene ontology annotations of cellular component, ribosome subunits compose ~12% of the total protein mass (Fig 2B), whereas membrane, mitochondrion, flagellum and nucleus proteins compose ~5.7, 5.3, 3.8, and 4.1% of the proteome, respectively (Fig 2B). The abundance of ribosomal subunits in our dataset is in agreement with the high activity of translation and synthesis of proteins observed in organisms in the logarithmic growth phase, such as the parasites used here (S3 Fig). In our dataset, ribosomal proteins of the large and small subunits consist of 64 and 36 proteins respectively. In agreement with the 1:1 stoichiometry, the average copy numbers of those proteins were $1.3 \pm 0.06 \times 10^5$ and $1.5 \pm 0.09 \times 10^5$, for the large and small subunits, respectively (Fig 2C).

## Energy metabolism molecules comprise about 8% of all molecules per cell

Quantitative proteomic data allowed us to determine the abundances of proteins involved in the main metabolic processes for energy production in those parasites. In our dataset, both glycosomal and cytosolic glycolytic enzymes were detected and quantified, representing near 1.2 million molecules per cell (Fig 2D, S2 Table). In addition, oxidative phosphorylation enzymes were represented by near 1.1 million molecules per cell. It is known that generation of ATP in promastigotes involves mainly glycolysis and oxidative phosphorylation [48]. In agreement, our dataset reveled that abundance of the proteins involved in those metabolic processes represents 3.7% of all protein molecules per cell. We also observed an important contribution of enzymes from other processes involved in ATP generation such as tricarboxylic acid cycle with an estimate of ~1.0 million molecules per cell, as well as amino acid oxidation and fatty

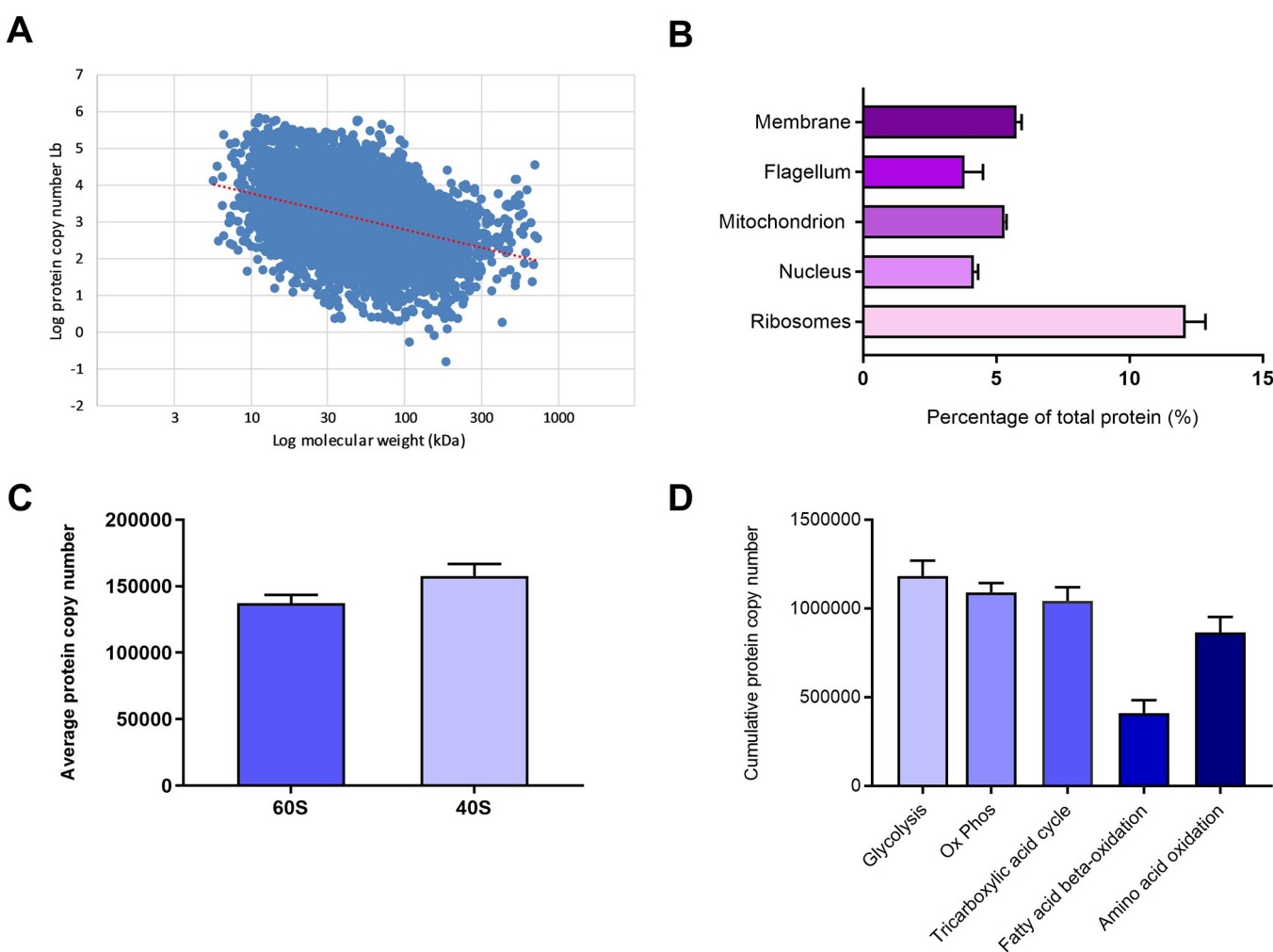

**Fig 2. Quantitative analysis of the proteomes of *Leishmania* (*Viannia*) species. (A)** Correlation between protein copy numbers and molecular weight, $R^2$ = 0.129. Every blue dot represents a single protein. Red dotted line = tendency line. **(B)** Protein abundance of basic components of promastigotes architecture based on the current gene ontology annotations of cellular component. The values represent the mean of percent of total cellular protein content ± SD (n = 14). **(C)** Abundance of ribosomal proteins of the large and small subunits. The values represent the mean of protein copy numbers ± SD (n = 14). **(D)** Cumulative abundance of proteins involved in the major energy metabolic processes. Bars show the total sum of the number of copies of proteins involved in each metabolic pathway ± SD (n = 14).

acid beta-oxidation, comprising an estimate of 860.000, and 410.000 molecules per cell, respectively (Fig 2D). Together, all those energy metabolic processes comprise ~ 4.6 million of molecules per cell, which corresponds to about 7.4% of all molecules per promastigote (Fig 2D, S2 Table).

## *Leishmania braziliensis*, *L. panamensis* and *L. guyanensis* exhibit significant differences in protein abundance

Statistical significance of differences in protein abundance among the species was determined by *t*-test analyses. In total, at FDR of 1%, the concentrations of 2112 proteins were significantly different between *L. braziliensis* and *L. panamensis*, 1639 between *L. braziliensis* and *L. guyanensis* and 1089 between *L. panamensis* and *L. guyanensis* (Fig 3A, S4 Table). Remarkably, the concentration of 185 proteins was different across the three species (Fig 3A), and near 45% of

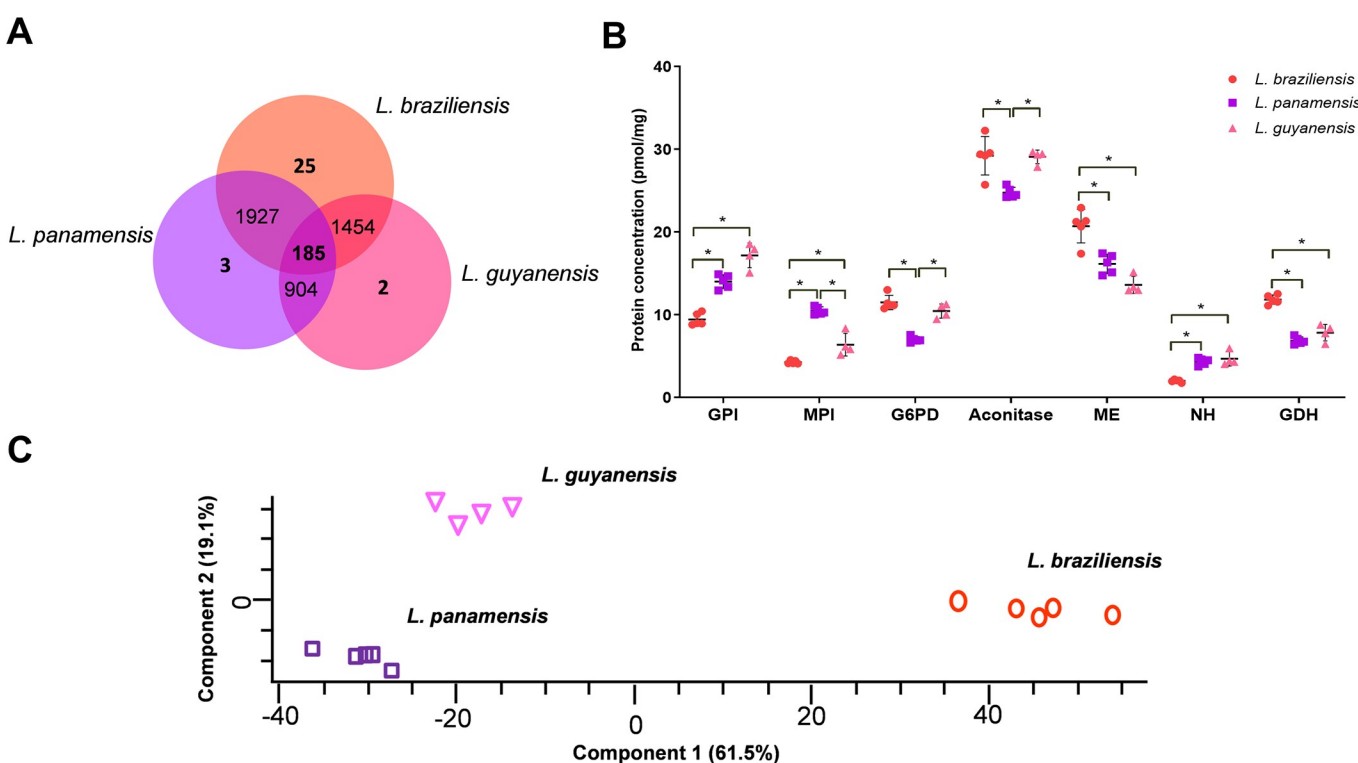

**Fig 3. Differences in protein abundance in the proteomes of *Leishmania* (*Viannia*) species. (A)** Venn diagram shows the number of differentially abundant proteins among *L. braziliensis*, *L. panamensis* and *L. guyanensis*. **(B)** Differential protein concentration of diverse isoenzymes used for *Leishmania* species identification (n = 5 biological replicates for *L. braziliensis* and *L. panamensis* and n = 4 for *L. guyanensis*). Asterisks denote significant differences in the proteomics dataset (FDR < 0.01). **(C)** Principal component analysis of the concentration values of enzymes of the main energy metabolic pathways including glycolysis, TCA, oxidative phosphorylation, fatty acid beta-oxidation and amino acid oxidation (S4 Table).

them are annotated as *uncharacterized*. Despite that, with the integration of omics data, the "*uncharacterized*" universe should represent a rich source of information in the future. Gene expression levels were analyzed by qPCR for 10 genes encoding for proteins that presented statistical differences of abundance among at least two species. We observed low correlation between mRNA levels assessed by qPCR and protein abundance (S4 Fig). This result is not unexpected since lack of correlation between transcript and protein levels is commonly observed both in prokaryotes and eukaryotes, including, *E. coli*, *Saccharomyces cerevisiae*, *Leishmania* spp. and mammalian cells [49–52]. In fact, in *Leishmania* spp. there is a constitutive expression of most of the genome and gene regulation mainly occurs at posttranscriptional level by differences on mRNA maturation and stability [53,54]. However, we are aware that our comparison was limited to 10 genes and we cannot rule out the possibility that deep pairwise comparison between transcriptomics and proteomics data could reveal potential correlations among mRNA and protein levels in *Leishmania*, at least for some genes in the strains studied here.

To analyze if there were particular groups of proteins that contributed the most for distinguishing the *L.* (*Viannia*) species, the 185 proteins were submitted to enrichment analysis. The *beta-alanine metabolism* (4.00-fold), *valine, leucine and isoleucine degradation* (3.73-fold) and *fatty acid degradation* (3.17-fold) were annotations of metabolic pathways significantly overrepresented. In addition, biological processes including *sterol biosynthetic process* (15-fold), *sulfur amino acid metabolic process* (11.9-fold), *monosaccharide metabolic process* (6.83-fold) *cellular amino acid metabolic process* (3.47-fold), *phospholipid biosynthetic process* (4.08-fold)

and *lipid metabolic process* (2.49-fold), among others, were also significantly enriched. In general, this analysis suggest that the abundance of proteins related to energy metabolism is distinct between the species here studied.

In order to know whether would have a chromosome that could contribute the most for those differences in protein abundance, we analyzed the chromosomal localization of the 185 proteins. With the exception of chromosomes 14 and 22, all the other chromosomes contribute with at least one differentially abundant protein. Interestingly, chromosomes 31, 35, 18 and 16 contributed with most of the differentially abundant proteins (16, 15, 14 and 13 proteins, respectively) (S5 Fig). Remarkably, proteins coded by genes located on that chromosomes are mainly transporters, peptidases and uncharacterized *integral components of membrane*, among others. Studies on *Leishmania* ploidy have revealed that chromosome copy number varies significantly between species, and even among strains, and that chromosome 31 is supernumerary in all species which ploidy state has been studied to date, including *L. braziliensis*, *L. guyanensis* and *L. panamensis* [7–16]. The ploidy of the chromosomes 35, 18 and 16 varies between disomic and trisomic states among different strains of *L. braziliensis*, *L. panamensis* and *L. guyanensis* [9,12,15,16]. Recently it was reported for *L. panamensis* that chromosomes 16 and 18 suffered changes in their somy after Sb-resistance selection [15, 16]. Whether the significant differences in protein abundance observed in our proteome dataset could be reflecting the genetic heterogeneity among those species, resulting from variation in chromosome number, remains to be clarified. However, the availability of in-depth quantitative protein data, as the one provided here, is a peremptory condition to answer that question.

Among those proteins that have abundancies significantly different among at least two species we identified HSP70, GP63, CPB, RNA pol II, DNA pol, CYB and even several of them have significantly different concentration among the three *L.* (*Viannia*) species studied here (S4 Table). Various enzymes such as malate dehydrogenase (MDH), isocitrate dehydrogenase (ICDH), glucose-6-phosphate dehydrogenase (G6PDH), 6-phosphogluconate dehydrogenase (6PGD) nucleoside hydrolase (NH), glutamate dehydrogenase (GDH), phosphoglucomutase (PGM), mannose phosphate isomerase (MPI), and glucose-6-phosphate isomerase (GPI) presented also significant differences in our dataset (S4 Table). All of those molecules are used for *L.* (*Viannia*) species typing using seminal approaches such as multilocus enzyme electrophoresis (MLEE) [55,56] or PCR variants including PCR-RFLP and multilocus sequence typing (MLST) [57–60]. Thus, our proteomics dataset shows that the differences in the abundance of those proteins between the *L. braziliensis*, *L. guyanensis* and *L. panamensis* strains analyzed here agree with the molecular polymorphisms commonly observed by MLEE, MLST and/or PCR-variant analyses. In addition, the depth of our data set allows us to recover the contribution of many other molecules not previously known for species differentiation.

### *Leishmania* (*Viannia*) species differ in the abundance of energy metabolism enzymes

Many of the isoenzymes used as gold standards for *Leishmania* species identification [55,56,61] presented significant differential abundance in our proteome dataset (Fig 3B). Since some of them are essential enzymes of the main energy metabolic pathways we decided to analyzed if would be possible to differentiate the three species relying only on the titers of enzymes of the energy metabolic pathways. The "energy profile" based on proteins concentration was first analyzed by PCA and we clearly observed three distinct clusters, each one corresponding to each species analyzed here (Fig 3C). Almost 80% of variability is explained by components 1 and 2, showing that there are clear differences in the energy metabolism among species. In addition, this result also shows that, in terms of energy metabolism, the *L. panamensis* and *L.*

*guyanensis* strains are more related together compared to the *L. braziliensis* one. This result is in agreement with the phylogenetic relationships deduced from different molecular approaches [55–58].

Further, we specifically compared the abundance of the proteins involved in glycolysis and oxidative phosphorylation, which are considered as the main metabolic pathways involved in energy generation in *Leishmania* promastigotes [20,48]. Remarkably, we observed statistical differences in the cumulative concentration of proteins involved in the glycolytic pathway among the species (Fig 4A and S4 Table). Such abundance is significantly higher in *L. braziliensis* (mean 513 pmol/mg) than in *L. panamensis* (mean 465 pmol/mg) or *L. guyanensis* (mean 473 pmol/mg), reinforcing the idea that these parasites seem to depend differentially on distinct energy substrates.

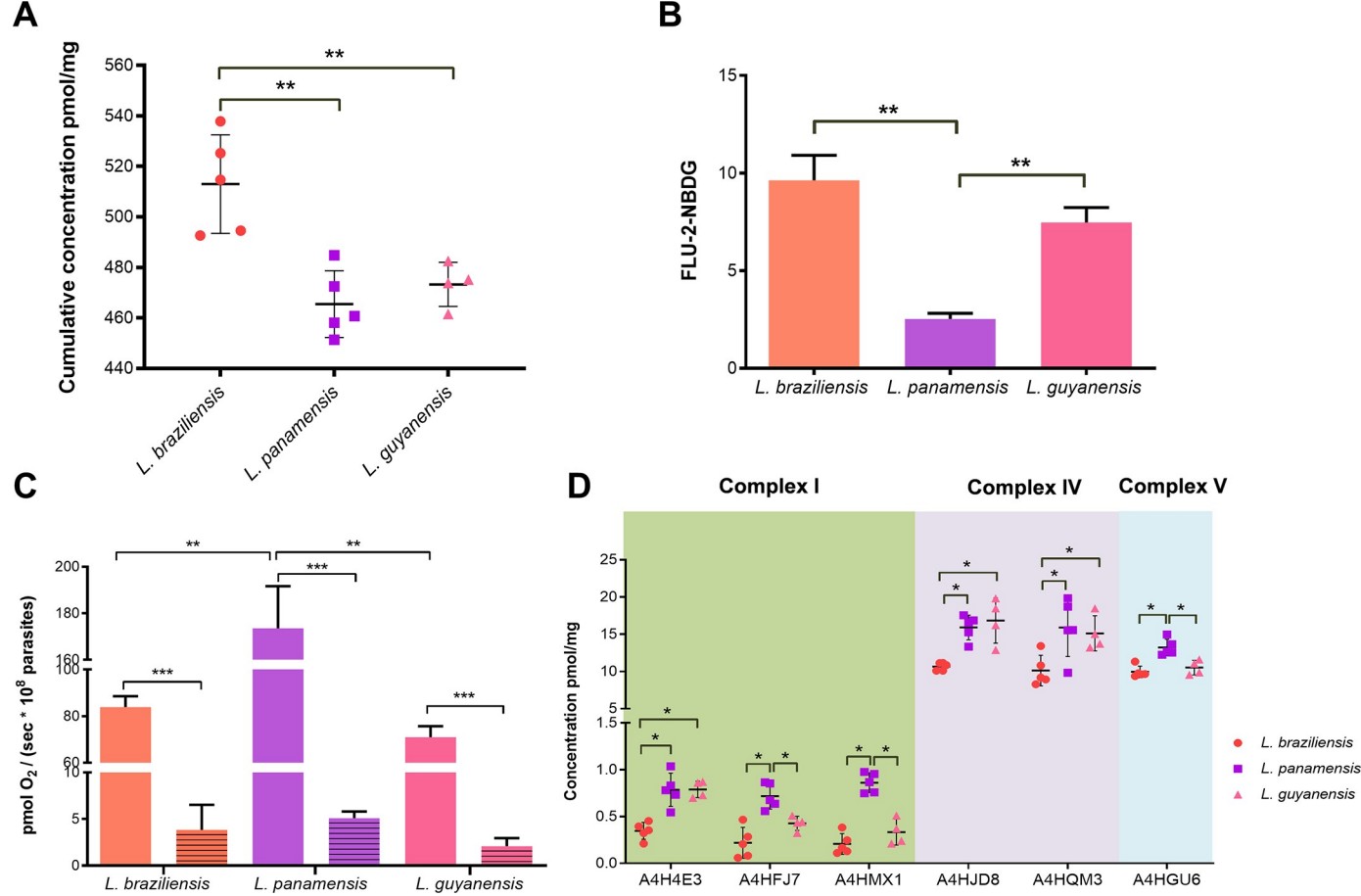

**Fig 4. *Leishmania* (*Viannia*) species significantly differ in the abundance of energy metabolism enzymes. (A)** Cumulative concentration of the proteins involved in glycolysis. Each symbol (dots, squares or triangles) show the total sum of the concentration values of proteins involved in glycolysis in each of the 5 biological replicates for *L. braziliensis* and *L. panamensis* or 4 biological replicates for *L. guyanensis*. Statistical differences by *t* test (p<0.003 Lb x Lp; p<0.006 Lb x Lg). **(B)** 2-NBDG uptake levels. Promastigotes were incubated with 300 μM 2-NBDG, a fluorescent glucose analogue, and further analysed by flow cytometry. Bars show mean ± SD of at least three independent experiments. Statistical differences by *t* test (p<0.001). **(C)** Oxygen consumption under routine conditions (open bars) or in the presence of 2 μM antimycin A (residual oxygen consumption-ROX, filled bars). ROX control shows a reduction up to 95% in mitochondrial functionality. Bars show mean and standard deviations of at least three independent experiments. Significant differences by *t* test. ** p<0.001; *** p<0.0001. **(D)** Concentrations of specific proteins involved in oxidative phosphorylation. Complex I (green background), complex IV (purple background) and complex V (blue background) enzymes are identified with the UniProt code (S4 Table). Values represent mean ± SD of protein concentration (pmol/mg) [n = 5 biological replicates for *L. braziliensis* (dots) and *L. panamensis* (squares) and n = 4 for *L. guyanensis* (triangles)]. *Statistical differences according to Perseus analysis (FDR < 0.01) and further analyzed by the Holm-Sidak method (p<0.01).

This result prompted us to analyze the levels of glucose uptake by each species through the evaluation of 2-NBDG consumption, a fluorescent glucose analogue [62]. Confirming our proteomics data, *L. braziliensis* exhibit significant 3.7-fold higher 2-NBDG uptake than *L. panamensis* (Fig 4B). In addition, this assay revealed that *L. guyanensis* also uptakes higher levels of 2-NBDG than *L. panamensis* (2.9-fold). *L. guyanensis* showed a lower 2-NBDG consumption level than *L. braziliensis*, although the difference was not significant (Fig 4B). Conversely, analysis of mitochondrial $O_2$ consumption showed that *L. panamensis* strain consumes 2-fold and 2.5-fold more $O_2$ than *L. braziliensis* and *L. guyanensis*, respectively (Fig 4C). In agreement with that results, proteomics data revealed significant differences in the abundance of enzymes involved in oxidative phosphorylation. Enzymes of the complex I (NADH dehydrogenase), complex IV (cytochrome c oxidase) and complex V (ATP synthase) exhibited higher concentration in *L. panamensis* than in the other two species (Fig 4D). In addition, in *L. guyanensis* we observed significantly higher titers for complex I and complex IV enzymes compared to *L. braziliensis* (Fig 4D). Taking together, our results suggest that *L. (Viannia)* species differ in the ways of ATP production. In addition, the analysis of the cumulative concentration of the enzymes from the other main energy pathways confirmed that both *L. guyanensis* and *L. panamensis* rely more on other metabolic pathways such amino acid oxidation and fatty acid beta-oxidation to obtain ATP than *L. braziliensis* (S6 Fig). Our results are in good agreement with early works showing that different *Leishmania* species consume glucose at different rates and that glycolysis, mitochondrial respiration and amino acid oxidation are the main ATP sources for promastigotes, whereas fatty acid beta-oxidation is the main source for amastigotes [48,63,64]. Further analysis of sugar uptake and oxygen consumption should be performed over the course of the parasites' growth curve in order to obtain indisputable data on the metabolic traits of each *L. (Viannia)* species during parasite differentiation. Although parasites change their metabolic profile during differentiation from procyclic to metacyclic stages [65, 66], the differences between species observed here are not related with a discrepancy in the rate of differentiation of parasites because: (i) all replicates of each species were collected at the same point of the growth curve (late log phase, S3 Fig); (ii) proteins such as SHERP/HASPB and META1, used as differentiation markers because their abundance increases highly in metacyclic parasites (stationary phase) [67–70], were not detected in our dataset; (iii) proteins involved in translation and synthesis of proteins, which are processes very active in organisms in the log phase (*i.e.*, dividing promastigotes) were highly abundant in our dataset; and (iv) differences in the protein abundance of ribosomal subunits, which are decreased in non-dividing metacyclic parasites (stat phase) [71], were not observed among the species studied here. Together, our data support the proposition that the three species of *L. (Viannia)* studied here have consistent differences in their energy metabolism. Our findings are in agreement with previous studies on the metabolome of different *L. (Leishmania)* species that demonstrated that *L. donovani*, *L. mexicana* and *L. major* are quite distinct in their metabolism [72]. Remarkably, large differences in consumption of amino acids were observed between those species [72]. Thus, potentially, the differences in energy metabolism sources could be a common trait in *Leishmania* spp.

Differences on energy metabolism probably reflect the different invertebrate hosts involved in transmission of each parasite species and their variable feeding sources. Indeed, the geographical origin of the *L. braziliensis* strain used here is the Atlantic forest at Brazilian' Bahia state where parasites can be transmitted by *Lutzomyia whitmani* and *Lu. intermedia*, whereas *L. panamensis* comes from the Colombian Andean mountain's region where the main vector is *Lu. gomezi*, and *L. guyanensis* strain was isolated in the Amazon region where it is mainly transmitted by *Lu. umbratilis* [73–75]. Such distinct landscapes exhibit different flora and fauna for sandflies feeding and those distinct ecological scenarios should influence the

preference for a particular metabolic pathway in parasites. The study of more strains of each species will corroborate whether those remarkably differences on energy metabolism are species-specific traits.

In addition to the differences in metabolism, we observed that concentration values of many proteins annotated as integral components of membrane, including transporters, surface antigens-like proteins, plasma membrane ATPases, and GP63, as well as proteins related to the metabolism of plasma membrane' lipids were significantly different across the three *L.* (*Viannia*) species (S4 Table, S7 Fig) and allowed distinguishing the three species studied here. Together, our results reveal a species-specific pattern of protein abundances that might be contributing to specific host adaptation and differentiation. Such remarkable differences could be exploited for better and suitable diagnosis tools. The mass spectrometry proteomics data have been deposited to the ProteomeXchange Consortium via the PRIDE [76] partner repository with the dataset identifier PXD017696.

## Conclusions

We provide the first in-depth quantitative proteomic analysis of *L. braziliensis*, *L. panamensis* and *L. guyanensis*, the etiological agents of cutaneous leishmaniasis in the Americas. Comprehensive comparison of the proteomes of strains representing these species allowed us to show the extension of the differences among those close related species and how such differences shape specie-specific traits. Our results clearly show that deep quantitative proteomics analysis is able to distinguish the three species, in a similar way that deep genome sequencing allows such distinction. Regarding that, we provided information of the total protein content of those species, their main cellular components, and differences in the abundance of specific proteins and energy metabolic pathways. Together our results indicate that differences observed at the major energy metabolic processes, among others, could be contributing for species-specific fitness. Notably, we accomplished that without laborious and expensive protocols for sample preparation and absolute protein quantification. Indeed, the TPA and proteomic ruler methods allowed the determination of absolute protein concentrations and copy numbers without the introduction of peptide or protein modifications. Remarkably, the rich quantitative data here provided is available for exploring new diagnostic systems that hopefully can be translated in affordable tools for using in the field.

## Supporting information

**S1 Fig. Peptide coverage and protein abundance correlation among replicates.** (A) 95% proteins were identified with at least three peptides. Profiles of Pearson's correlation coefficient of protein abundance among (B) *L. braziliensis* replicates, (C) *L. panamensis* replicates, (D) *L. guyanensis* replicates. Plotted values represent the total protein fraction of each individual protein calculated by the *total protein approach*.
(TIF)

**S2 Fig. Functional annotations of protein identified in *L.* (*Viannia*) species proteome.** (A) Number and percentage of proteins in our dataset with any functional annotation for the main gene ontology (GO) categories of Cellular component, Molecular function and Biological process. (B) Number of proteins identified in each one of the main cellular components of *Leishmania* promastigotes, according to GO annotation for cellular component.
(TIF)

**S3 Fig. Growth curves of *L. braziliensis*, *L. panamensis* and *L. guyanensis*.** Every dot, square or triangle represents mean ± SD of biological triplicates for *L. braziliensis*, *L. panamensis* and

*L. guyanensis*, respectively. Growth curves started by adding 1 x 10$^5$ parasites/mL in 10 mL of Schneider's medium supplemented with 10% of FBS. Parasites proliferation was evaluated every 24 h during two weeks by optical microscopy using a hemocytometer.
(TIF)

**S4 Fig. Protein abundance and corresponding mRNA levels for 10 selected genes.** Values represent mean ± SD of protein concentration (pmol/mg) [n = 5 biological replicates for *L. braziliensis* (dots) and *L. panamensis* (squares) and n = 4 for *L. guyanensis* (triangles)]. * Statistical differences according to Perseus analysis (FDR < 0.01). Bars represent mean ± SD of mRNA relative levels determined by qPCR. * Significant differences by *t* test.
(TIF)

**S5 Fig. Chromosomal localization of differentially abundant proteins among the three species.**
(TIF)

**S6 Fig. Cumulative concentration of the proteins involved in other energy metabolic pathways.** Bars show the mean of total sum of the concentration values in each species (n = 5 biological replicates for *L. braziliensis* and *L. panamensis* and n = 4 for *L. guyanensis*). Statistical differences by *t* test (p<0.01 Lb x Lb; p<0.005 Lb x Lg; p<0.0001 Lp x Lg).
(TIF)

**S7 Fig. Proteins involved in lipids metabolism show significant differences in abundance.** (A) Concentration values of different plasma membrane transporters. (B) Concentration values of different plasma membrane proteins. (C) Concentration values of proteins involved in biosynthesis/metabolism of sphingolipid (SL), phospholipid (PL) (green background), isoprenoid and sterols (blue background) and other lipids (purple background). Values represent mean of pmol/mg ± SD (n = 5 biological replicates for *L. braziliensis* and *L. panamensis* and n = 4 for *L. guyanensis*). All plotted values were statistically different across the three species in our proteomics dataset (FDR < 0.01) (S4 Table).
(TIF)

**S1 Table. Primer sequences used for qPCR amplification.**
(XLSX)

**S2 Table. Protein groups identification and histone ruler.**
(XLSX)

**S3 Table. Reproducibility among biological replicates.**
(XLSX)

**S4 Table. Statistical significance of differences in protein abundance among the species.**
(XLSX)

## Acknowledgments

The authors are grateful to Prof. Matthias Mann for continuous support. We thank Katharina Zettl for technical help with mass spectrometric measurements, and Dr. Rosane Temporal -quality manager of LPL-FIOCRUZ-RJ and all the staff of CLIOC for the technical assistance and quality advising.

## Author Contributions

**Conceptualization:** Patricia Cuervo.

**Data curation:** Nathalia Pinho, Patricia Cuervo.

**Formal analysis:** Jacek R. Wiśniewski, Geovane Dias-Lopes, Leonardo Saboia-Vahia, Jose Batista de Jesus, Gabriel Padrón, Patricia Cuervo.

**Funding acquisition:** Elisa Cupolillo, Jose Batista de Jesus, Patricia Cuervo.

**Investigation:** Nathalia Pinho, Jacek R. Wiśniewski, Geovane Dias-Lopes, Leonardo Saboia-Vahia, Ana Cristina Souza Bombaça, Camila Mesquita-Rodrigues, Rubem Menna-Barreto, Gabriel Padrón, Patricia Cuervo.

**Methodology:** Jacek R. Wiśniewski.

**Project administration:** Patricia Cuervo.

**Resources:** Jacek R. Wiśniewski, Rubem Menna-Barreto, Elisa Cupolillo, Jose Batista de Jesus, Patricia Cuervo.

**Supervision:** Patricia Cuervo.

**Validation:** Ana Cristina Souza Bombaça, Gabriel Padrón.

**Visualization:** Nathalia Pinho, Patricia Cuervo.

**Writing – original draft:** Nathalia Pinho, Patricia Cuervo.

**Writing – review & editing:** Jacek R. Wiśniewski, Ana Cristina Souza Bombaça, Elisa Cupolillo, Jose Batista de Jesus, Gabriel Padrón, Patricia Cuervo.

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
