## [Decision Letter · Decision Letter 0]

13 Apr 2020

Dear Dr. Cuervo,

Thank you very much for submitting your manuscript "In-depth quantitative proteomics uncovers specie-specific metabolic programs in Leishmania (Viannia) species" for consideration at PLOS Neglected Tropical Diseases. As with all papers reviewed by the journal, your manuscript was reviewed by members of the editorial board and by several independent reviewers. In light of the reviews (below this email), we would like to invite the resubmission of a significantly-revised version that takes into account the reviewers' comments. 

We cannot make any decision about publication until we have seen the revised manuscript and your response to the reviewers' comments. Your revised manuscript is also likely to be sent to reviewers for further evaluation.

Sincerely,

Armando Jardim, PhD

Associate Editor

Walderez Dutra

Deputy Editor

Reviewer's Responses to Questions

**Key Review Criteria Required for Acceptance?**

**Methods**

-Are the objectives of the study clearly articulated with a clear testable hypothesis stated?

-Is the study design appropriate to address the stated objectives?

-Is the population clearly described and appropriate for the hypothesis being tested?

-Is the sample size sufficient to ensure adequate power to address the hypothesis being tested?

-Were correct statistical analysis used to support conclusions?

-Are there concerns about ethical or regulatory requirements being met?

Reviewer #1: The manuscript by Patricia Cuervo’s group utilizes a label free quantitative proteomic approach to map the ultra-deep and to compare the proteomes of Leishmania (Viannia) species namely L. braziliensis, L. panamensis and L. guyanensis. The authors mapped more than 80% of parasite proteins using a multi-enzyme digestion filter aided sample preparation (FASP) method and coupled with analysis using a high accuracy Orbitrap Q-Exactive HF mass spectrometer. The authors have successfully applied previously established methods such as total protein approach (TPA) to quantitate abundance of proteins based on spectral intensities and histone ruler approach to estimate protein copy number per Leishmania cell. As indicated in the title, the authors report on differences in energy metabolism precursors between the three species. While L. braziliensis rely on glycolysis, the other two utilize mitochondrial metabolites as source for metabolic energy.

Reviewer #2: All analyzes are well conducted but the study design provides major flaws that challenge the main conclusions drawn in this manuscript as detailed below.

Reviewer #3: see comment 1 under editorial and data representation modifications .

**Results**

-Does the analysis presented match the analysis plan?

-Are the results clearly and completely presented?

-Are the figures (Tables, Images) of sufficient quality for clarity?

Reviewer #1: This is a very nice paper. Its major contribution is the new and in-depth proteomic analysis of Leishmania (Viannia) species. Significantly, the data this study provides (about 7000 proteins per cell and their absolute abundance values) is significantly more comprehensive than in previous publications. This database justifies publication in PLoS NTD. 

This is a very nice paper. Its major contribution is the new and in-depth proteomic analysis of Leishmania (Viannia) species. Significantly, the data this study provides (about 7000 proteins per cell and their absolute abundance values) is significantly more comprehensive than in previous publications. This database justifies publication in PLoS NTD. 

The major weakness of this study is the biology behind the numbers. The authors sampled late log phase parasites (Materials and Methods). This is a border phase that require serious care to make sure that all the biological repeats were harvested at exactly same conditions. For example, the change in the abundance in metabolic enzymes led the authors to conclude on the differences in the source of metabolic energy. In 1974, Mukkada et al. (Mukkada et al., 1974) and Opperdoes group 23 years later (Ter Kuile and Opperdoes, 1992) showed that log phase promastigotes utilize glucose as source of energy and when they reach stationary phase they move to proline and alanine, i.e. Kerb's cycle intermediates. It is likely that the observation made by the authors of this paper were due to timing of harvesting rather than real metabolic specificities. The authors must provide a strong evodence that all three strains with their biological repeats were harvested at identical culture conditions. In mid log or stationary phases conditions are much more stable than late log (If I were to choose the experimental conditions, I would pick mid log phase). The authors should address this or avoid getting into conclusion on comparative metabolism. 

I have a few additional major and minor comments:

1. The title should be reconsidered. 

2. Abstract lines 32-33: "…Leishmania proteome remained mostly uncharacterized" and line 86 in Introduction. These statements are simply not true. Over 20 research papers have been published on Leishmania proteomics, each served its purpose and most of them used state-of-the-Art technologies. Together they are being used by the community. I assume that in 3 or 5 years someone else will use an advanced method that will undermine this "advanced" analysis. Modesty will be more appropriate here. 

3. Introduction line 50 "…species-specific differentiation…" did you mean "differences?

4. Methods lines 131-132: see the paragraph above. Indicate how did you make sure parasites are all at exactly same growth phase conditions.

5. The authors have not provided the complete peptide list for each of the three Leishmania (Viannia) species. This is one of the first shotgun proteomic studies I have come across in many years where the authors have not provided peptide information as a part of the manuscript.

6. If one looks at supplementary table 2 it is not possible to identify a single L. panamensis or L. guyanensis accession from UniProt database. However, in the methods section proteomic data analysis the authors state that "The data were analyzed with MaxQuant Software using the L. braziliensis, L. guyanensis and L. panamensis protein sequences available at UniProt database." It is not very clear what database dependent searches have been undertaken by the authors and this bring to question the veracity of study and identifications. The authors should make clear if they have searched the MS data from all the three Leishmania species only against a L. braziliensis database? Or they have searched a combined uniprot database of the three species, in that case why only L. braziliensis UniProt Ids are shown in the supplementary data and not the other two species.

7. Continuing with the above question have the authors looked at species specific genes (which would be handful) identified based on genome sequencing data which are truly unique to each L. braziliensis, L. guyanensis and L. panamensis species used in the study. These genes would make each of species different from one another. This shouldn’t be mistaken for the proteomic expression analysis that the authors undertook where they have identified 25 (L. braziliensis), 2 (L. guyanensis) and 3 (L. panamensis) species specific protein expression as shown in Figure 3A.

8. In the results section: How do the authors differentiate between close paralog which are more than 95% identical and peptide assignments. The authors seem to be overestimating their protein identifications. If they are suggesting that each paralog in a multi-gene family are translated equally considering Leishmania gene expression is regulated post-transcriptionally. They should clarify this aspect of in case of multi-gene families or genes with more than one paralog.

9. The authors should provide a complete list of genes including gene ids which have one or more paralogs in each of the Leishmania species they have used in their analysis and the peptides identified in the present study that map to or are shared across the different paralogs in each Leishmania species.

10. On line 269 the authors state “our results show that global protein abundances obtained by deep proteomics are able to clearly distinguish the three species, in a similar way that deep genome sequencing allows such distinction [9,12].” What are the species specific proteins have been identified by the authors, which are unique to just one species but not identified. The authors should describe one section in the results and discussion regarding these species-specific proteins identified in the study. These can be the ones that make one species of Leishmania have the specific pathogenicity not found in the other species. Considering more than 98% of genes are conserved across the old and new world species this should not be a difficult task for the authors.

11. In Figure 5 the authors should provide density plots for metallopeptidase activity bands in the zymographic analysis for the three species.

12. For the manuscript whose sole claim to glory is the depth of the protein identifications. The authors have provided very poor images which appear grainy at the best and pixelated at worst. The authors should re-export higher dpi images from MaxQuant or Perseus software and provide it again.

13. The authors have submitted their data to Proteomexchange repository. However, the data is not made publicly available. Even as a reviewer I am not able to access the data. The authors should make their data in Proteomexchange accessible to public or should provide access to the reviewers. Hopefully they have provided processed searched data files and also raw data files as part submission to Proteome exchange if not they should do it. 

14. The results and discussion section of the manuscript is very lengthy, exhaustive and at times it quite mind-numbing to read such a lengthy manuscript. The authors should shorten the results and discussion section of the manuscript and only describe the most important finding in a succinct manner. 

15. Figure legends and tables are smack in the middle of the manuscript which disturbs the flow. The authors should know that at later time point the manuscript once accepted is typeset. So, the figures and tables could be having been provided at the end of the manuscript

Reviewer #2: All data are clearly presented in figures and tables, and properly described in the text.

Reviewer #3: see comment 2 and 3 in editorial and data representation modifications section.

**Conclusions**

-Are the conclusions supported by the data presented?

-Are the limitations of analysis clearly described?

-Do the authors discuss how these data can be helpful to advance our understanding of the topic under study?

-Is public health relevance addressed?

Reviewer #1: See above

Reviewer #2: As detailed below, the study has two main flaws that challenge the major findings on (i) comparative analysis between parasite species based on absolute quantification and (ii) the identification of species-specific differences in energy metabolism.

Reviewer #3: see general comments.

**Editorial and Data Presentation Modifications?**

Reviewer #1: Peptide list should be free to the community. See coment 5

Reviewer #2: (No Response)

Reviewer #3: 1. Differences in abundance of metabolic enzymes in the parasites is clearly dependent on differentiation. In the chosen set up this means, it is critically dependent on the growth characteristics of the chosen strains. Thus, I would expect authors to analyse, or at least stress that for conclusive data, the sugar uptake and oxygen consumption measurements should be performed over the course of a full growth curve in vitro for all the isolates. 

2. GP63 is notoriously showing lower than expected abundance in proteomic data sets of the parasites and this may relate to solubilisation problems since also in their approach lysates are cleared before processing them further for eventual peptide determination by MS. Such a technical explanation for inconsistencies is neither considered nor discussed. As is the general problem whether e.g. membrane proteins are indeed as quantitatively sampled as soluble proteins or become systematically under-represented in such data sets. This should be discussed. 

3. Data and text regarding figures 5-7 could be either moved to the supplementary section or left out altogether, since they simply illustrate ideas on how to develop hypothesis that try to mine and exploit the really great proteome dataset. I would suggest leaving this exercise to the interested reader and scientific community to which this resource is becoming available. Also, this will leave time to substantiate the authors’ own hypothesis without presenting examples that are not yet revealing. 

4. A major aspect of the rules governing protein abundance in these parasites has been translational bias in codon usage (e.g. see Jeacock et al. Elife. 2018 Mar 15;7. pii: e32496. doi: 10.7554/eLife.32496). It seems that the presented datasets lend themselves perfectly to analyse this and, possibly, this can qualify also the answer to comment 2 above.

**Summary and General Comments**

Reviewer #1: The manuscript by Patricia Cuervo’s group utilizes a label free quantitative proteomic approach to map the ultra-deep and to compare the proteomes of Leishmania (Viannia) species namely L. braziliensis, L. panamensis and L. guyanensis. The authors mapped more than 80% of parasite proteins using a multi-enzyme digestion filter aided sample preparation (FASP) method and coupled with analysis using a high accuracy Orbitrap Q-Exactive HF mass spectrometer. The authors have successfully applied previously established methods such as total protein approach (TPA) to quantitate abundance of proteins based on spectral intensities and histone ruler approach to estimate protein copy number per Leishmania cell. As indicated in the title, the authors report on differences in energy metabolism precursors between the three species. While L. braziliensis rely on glycolysis, the other two utilize mitochondrial metabolites as source for metabolic energy. 

This is a very nice paper. Its major contribution is the new and in-depth proteomic analysis of Leishmania (Viannia) species. Significantly, the data this study provides (about 7000 proteins per cell and their absolute abundance values) is significantly more comprehensive than in previous publications. This database justifies publication in PLoS NTD. 

The major weakness of this study is the biology behind the numbers. The authors sampled late log phase parasites (Materials and Methods). This is a border phase that require serious care to make sure that all the biological repeats were harvested at exactly same conditions. For example, the change in the abundance in metabolic enzymes led the authors to conclude on the differences in the source of metabolic energy. In 1974, Mukkada et al. (Mukkada et al., 1974) and Opperdoes group 23 years later (Ter Kuile and Opperdoes, 1992) showed that log phase promastigotes utilize glucose as source of energy and when they reach stationary phase they move to proline and alanine, i.e. Kerb's cycle intermediates. It is likely that the observation made by the authors of this paper were due to timing of harvesting rather than real metabolic specificities. The authors must provide a strong evodence that all three strains with their biological repeats were harvested at identical culture conditions. In mid log or stationary phases conditions are much more stable than late log (If I were to choose the experimental conditions, I would pick mid log phase). The authors should address this or avoid getting into conclusion on comparative metabolism. 

I have a few additional major and minor comments:

1. The title should be reconsidered. 

2. Abstract lines 32-33: "…Leishmania proteome remained mostly uncharacterized" and line 86 in Introduction. These statements are simply not true. Over 20 research papers have been published on Leishmania proteomics, each served its purpose and most of them used state-of-the-Art technologies. Together they are being used by the community. I assume that in 3 or 5 years someone else will use an advanced method that will undermine this "advanced" analysis. Modesty will be more appropriate here. 

3. Introduction line 50 "…species-specific differentiation…" did you mean "differences?

4. Methods lines 131-132: see the paragraph above. Indicate how did you make sure parasites are all at exactly same growth phase conditions.

5. The authors have not provided the complete peptide list for each of the three Leishmania (Viannia) species. This is one of the first shotgun proteomic studies I have come across in many years where the authors have not provided peptide information as a part of the manuscript.

6. If one looks at supplementary table 2 it is not possible to identify a single L. panamensis or L. guyanensis accession from UniProt database. However, in the methods section proteomic data analysis the authors state that "The data were analyzed with MaxQuant Software using the L. braziliensis, L. guyanensis and L. panamensis protein sequences available at UniProt database." It is not very clear what database dependent searches have been undertaken by the authors and this bring to question the veracity of study and identifications. The authors should make clear if they have searched the MS data from all the three Leishmania species only against a L. braziliensis database? Or they have searched a combined uniprot database of the three species, in that case why only L. braziliensis UniProt Ids are shown in the supplementary data and not the other two species.

7. Continuing with the above question have the authors looked at species specific genes (which would be handful) identified based on genome sequencing data which are truly unique to each L. braziliensis, L. guyanensis and L. panamensis species used in the study. These genes would make each of species different from one another. This shouldn’t be mistaken for the proteomic expression analysis that the authors undertook where they have identified 25 (L. braziliensis), 2 (L. guyanensis) and 3 (L. panamensis) species specific protein expression as shown in Figure 3A.

8. In the results section: How do the authors differentiate between close paralog which are more than 95% identical and peptide assignments. The authors seem to be overestimating their protein identifications. If they are suggesting that each paralog in a multi-gene family are translated equally considering Leishmania gene expression is regulated post-transcriptionally. They should clarify this aspect of in case of multi-gene families or genes with more than one paralog.

9. The authors should provide a complete list of genes including gene ids which have one or more paralogs in each of the Leishmania species they have used in their analysis and the peptides identified in the present study that map to or are shared across the different paralogs in each Leishmania species.

10. On line 269 the authors state “our results show that global protein abundances obtained by deep proteomics are able to clearly distinguish the three species, in a similar way that deep genome sequencing allows such distinction [9,12].” What are the species specific proteins have been identified by the authors, which are unique to just one species but not identified. The authors should describe one section in the results and discussion regarding these species-specific proteins identified in the study. These can be the ones that make one species of Leishmania have the specific pathogenicity not found in the other species. Considering more than 98% of genes are conserved across the old and new world species this should not be a difficult task for the authors.

11. In Figure 5 the authors should provide density plots for metallopeptidase activity bands in the zymographic analysis for the three species.

12. For the manuscript whose sole claim to glory is the depth of the protein identifications. The authors have provided very poor images which appear grainy at the best and pixelated at worst. The authors should re-export higher dpi images from MaxQuant or Perseus software and provide it again.

13. The authors have submitted their data to Proteomexchange repository. However, the data is not made publicly available. Even as a reviewer I am not able to access the data. The authors should make their data in Proteomexchange accessible to public or should provide access to the reviewers. Hopefully they have provided processed searched data files and also raw data files as part submission to Proteome exchange if not they should do it. 

14. The results and discussion section of the manuscript is very lengthy, exhaustive and at times it quite mind-numbing to read such a lengthy manuscript. The authors should shorten the results and discussion section of the manuscript and only describe the most important finding in a succinct manner. 

15. Figure legends and tables are smack in the middle of the manuscript which disturbs the flow. The authors should know that at later time point the manuscript once accepted is typeset. So, the figures and tables could be having been provided at the end of the manuscript

Reviewer #2: The manuscript "In-depth quantitative proteomics uncovers specie-specific metabolic programs in Leishmania (Viannia) species" by Pinho et al. provides quantitative insight into the proteomes of three Leishmania strains, one each corresponding to L. braziliensis, L. panamensis and L. guyanensis species. The authors compare the results both by relative quantification on total protein levels, as well as by absolute quantification using histone expression as a biological ruler to calculate protein concentration and copy number on a cellular level. Based on these results, the authors investigate energy metabolism in these parasites and provide biochemical evidence that L. braziliensis energy production relies on glycolysis, while L. panamensis and L. guyanensis seem to depend mainly on mitochondrial respiration. The manuscript is well written and provides deep proteomic insight into three Leishmania species of medical importance. As such it is a new and very useful resource for the larger parasitology community. However, the study has two main flaws that challenges the major findings:

First, the absolute quantification is based on calculations assuming that the three parasite strains are disomic for all chromosomes and that histones are expressed equally across these strains. The frequent aneuploidies observed in parasites in culture and the triploid state of L. braziliensis (trisomic state of all chromosomes, with higher ploidies observed in few chromosomes, see Rogers et al, PMID: 22038252) calls these assumptions into question. Experimental proof of the copy number and expression levels for the histone(s) used for normalization is required, for example by HTseq or RTqPCR analysis to assess gene copy number, and Western Blot analysis of histone expression levels (various antibodies are available in the community). 

Second, even though the differences in energy metabolism between strains is potentially very interesting, the analysis of just a single strain per parasite species raises the question whether this observation is just a strain- rather than species-specific phenomenon. Indeed, genomic and phenotypic variability between isolates of a given species is one of the hallmarks of Leishmania. Thus, species-specific differences need to be validated by analysing at least three different strains (best of distinct geographic origin) per parasite species using the established biochemical approaches.

Reviewer #3: Pinho and colleagues report on their proteomic analysis of New World Leishmania species of the Viannia complex. The authors are commended for this work, in particular, for significantly expanding the proteomic database in this field. 

The strengths of the study are the application of a state of the art, label free quantitative approache to analyse the proteome of in vitro grown parasite isolates of L. braziliensis, L. guayanensis and L. panamensis. This is a significant, stand alone contribution to our knowledge base and contributes information on nearly 80% of the genomic information derived, predicted protein ensemble that makes up these parasites.

Weaknesses are mostly related to the mentioned issues and, in comparison to the strength, are considered minor.

PLOS authors have the option to publish the peer review history of their article (what does this mean?). If published, this will include your full peer review and any attached files.

Reviewer #1: No

Reviewer #2: No

Reviewer #3: No
---

## [Editor Report · Decision Letter 1]

22 Jun 2020

Dear Dr. Cuervo,

We are pleased to inform you that your manuscript 'In-depth quantitative proteomics uncovers specie-specific metabolic programs in Leishmania (Viannia) species' has been provisionally accepted for publication in PLOS Neglected Tropical Diseases.

Best regards,

Armando Jardim, PhD

Associate Editor

Walderez Dutra

Deputy Editor

---

## [Editor Report · Acceptance letter]

10 Aug 2020

Dear Dr. Cuervo,

We are delighted to inform you that your manuscript, "In-depth quantitative proteomics uncovers specie-specific metabolic programs in Leishmania (Viannia) species," has been formally accepted for publication in PLOS Neglected Tropical Diseases.

Best regards,

Shaden Kamhawi

co-Editor-in-Chief

Paul Brindley

co-Editor-in-Chief
